# Determining Appropriate Numbers and Times of Daily Measurements Using GreenFeed System to Estimate Ruminal Methane Emission of Meat Goats

**DOI:** 10.3390/ani14060835

**Published:** 2024-03-08

**Authors:** Dereje Tadesse, Ryszard Puchala, Hirut Yirga, Amlan Kumar Patra, Terry Allen Gipson, Byeng Ryel Min, Arthur Louis Goetsch

**Affiliations:** 1American Institute for Goat Research, Langston University, Langston, OK 73050, USA; dereje.gulich@langston.edu (D.T.); hirut.tejeji@langston.edu (H.Y.); amlan.patra@langston.edu (A.K.P.); terry.gipson@langston.edu (T.A.G.); arthur.goetsch@langston.edu (A.L.G.); 2Department of Animal Sciences, Debre Berhan University, Debre Berhan P.O. Box 445, Ethiopia; 3Military Institute of Hygiene and Epidemiology, Kozielska 4, 00-163 Warsaw, Poland; 4Department of Animal and Range Sciences, Haramaya University, Dire Dawa P.O. Box 138, Ethiopia; 5Department of Agricultural and Environmental Sciences, Tuskegee University, Tuskegee, AL 36088, USA; bmin1@tuskegee.edu

**Keywords:** enteric fermentation, greenhouse-gas emission, portable calorimetry, small ruminants

## Abstract

**Simple Summary:**

Methane emitted by ruminant livestock contributes to climate change and represents a considerable waste of energy. Several methane measurement devices have been developed to measure ruminal methane emission. The GreenFeed system (GFS) was introduced as a static short-term measurement device to quantify methane emission by groups of animals in pen or pasture settings. Currently, protocols are available for measuring methane emission by cattle using GFS, but none are available for goats. This study, therefore, was conducted to determine appropriate numbers (3, 4 and 6 times/day) and times of daily measurements to estimate methane emission by goats with the GFS. Also, methane emission determined with a calorimetry system (CS) was compared with that quantified with GFS. The frequency of daily measurements did not affect methane emission estimates with the GFS system, but methane emission with the GFS was much higher than with the CS. The relationship between CS and GFS, which entailed four- and six-times daily measurements, was stronger. The study suggests a protocol involving at least four daily measurements may be useful to detect treatment differences or animal ranking for methane emission. However, using the GFS for goats under the present conditions estimated higher methane emissions compared with the CS, and thus it requires modifications to improve methane-emission estimates.

**Abstract:**

The study was conducted to determine appropriate numbers and times of daily gas measurements to estimate total daily methane (CH_4_) emission of meat goats using a GreenFeed system (GFS). A replicated 4 (four measurement protocols) × 4 (four periods) Latin square design was employed with 16 Boer wethers in a confinement pen setting. Measurement protocols entailed three (G-3T; 0600–0700, 1400–1500, and 2200–2300 h), four (G-4T; 0700–0800, 1300–1400, 1900–2000, and 0100–0200 h), and six (G-6T; 0800–0900, 1200–1300, 1600–1700, 2000–2100, 0000–0100, and 0400–0500 h) times for daily measurement periods in GFS. The fourth protocol was continuous measurement over 24 h with animals in an open-circuit respiration calorimetry system (CS). Oat hay was given in individual feeders, and a small predetermined quantity of a pelleted concentrate supplement (bait) was dispensed by the GFS or manually offered for the CS. Overall, total dry matter (DM) intake (614, 625, 635, and 577 g/day for CS, G-3T, G-4T, and G-6T, respectively; SEM = 13.9) and digestible DM intake (359, 368, 374, and 320 CS, G-3T, G-4T, and G-6T, respectively; SEM = 15.9) were lower for CS than for G-3T, G-4T, and G-6T (*p* < 0.05), but these variables were not different among the GFS protocols. There was a significant (*p* < 0.001) effect of measurement protocol on CH_4_ emission in g/day (11.1, 25.6, 27.3, and 26.7 for CS, G-3T, G-4T, and G-6T, respectively; SEM = 1.11), g/kg DM intake (19.3, 46.4, 43.9, and 42.4 for CS, G-3T, G-4T, and G-6T, respectively; SEM = 2.03), and g/kg body weight (0.49, 1.11, 1.18, and 1.16 for CS, G-3T, G-4T, and G-6T, respectively; SEM = 0.052), with values being much lower for CS than for G-3T, G-4T and G-6T. Conversely, CH_4_ emission was similar among the GFS protocols despite differences in the time and number of daily visits (2.03, 2.76, and 3.75 visits for G-3T, G-4T, and G-6T, respectively; SEM = 0.114; *p* < 0.001). Pearson correlation (r) analysis indicated a moderate to high (*p* < 0.05) correlation between CS and G-3T (r = 0.62 for CH_4_ in g/day and r = 0.59 for CH_4_ in g/kg BW), CS and G-4T (r = 0.67 for CH_4_ in g/day and r = 0.76 for CH_4_ in g/kg BW), and CS and G-6T (r = 0.70 for CH_4_ in g/day and r = 0.75 for CH_4_ in g/kg BW). However, the correlation coefficient for CH_4_ in g/kg DM intake was low between CS and G-3T (r = 0.11) and CS and G-6T (r = 0.31) but slightly greater between CS and G-4T (r = 0.47). In conclusion, the results suggest that CH_4_ emissions using GFS in a confinement setting were greater compared with the CS in goats, but CH_4_-emission estimation using the GFS correlated with the CH_4_ emission in the CS system with a stronger relationship for the four times of daily measurements.

## 1. Introduction

Greenhouse-gas emissions are a crucial global concern due to their significant impact on climate change, causing increasing numbers of climate disasters in the USA [1]. Agricultural activities account for about 11.6% of total greenhouse-gas emissions (on a carbon dioxide-equivalent basis) globally [2]. Animal agriculture is responsible for 5.8% of total anthropogenic greenhouse-gas emissions, and CH_4_ and nitrous oxide are the primary greenhouse gases in livestock production, with enteric and manure CH_4_ representing 32% of total anthropogenic CH_4_ emissions [2]. According to the Intergovernmental Panel on Climate Change [3], sheep and goats collectively contribute approximately 6.5% of global emissions of CH_4_, with goats accounting for 41% of this share. Beyond its prominent role in the greenhouse-gas effect, enteric-CH_4_ emissions represent a sizeable dietary energy loss of approximately 2–12% in cattle [4], 1.8 to 11% in sheep [5], and 2 to 9% in goats [6]. To address both the challenges of global warming and the inefficiencies in dietary energy utilization, the scientific community has explored and suggested various mitigation strategies, including interventions of diets, animals, management, ruminal fermentation, and microbial metabolism [7,8]. Simultaneously, concerted efforts have been made to accurately quantify enteric-CH_4_ emissions by ruminants to prepare a greenhouse-gas inventory from agricultural sources and the evaluation of different mitigation strategies. 

Currently, several techniques are available for measuring CH_4_ emission by ruminant animals, notably, using respiration calorimeter and sulfur hexafluoride (SF_6_) methods [9]. Respiration chambers are regarded as a ‘gold’ standard method due to their greater precision and reproducibility of gas measurements [10]. The respiration chamber technique, however, is associated with high costs, a significant disruption to normal animal behavior and limited representativeness of natural environments [10,11,12]. The SF_6_ technique for CH_4_ measurement can be employed in natural grazing conditions, but this method incurs greater variability in CH_4_ estimates and lower reproducibility compared with the respiration chamber technique [10,12]. A recent innovation, the GreenFeed system (GFS), introduced by C-Lock Inc. (Rapid City, SD, USA), offers a novel approach to estimating CH_4_ emissions in ruminants in various settings, including indoor and outdoor group housing or grazing. In contrast to traditional methods, CH_4_ measurement with GFS is based on regular and voluntary short-term visits by animals to the GFS [13]. Numerous studies have been conducted to assess accuracy of GFS in estimating CH_4_ emission by beef and dairy cattle (e.g., Hammond et al. [12,13]; Huhtanen et al. [14]; Arbre et al. [15]; Jonker et al. [16]; Doreau et al. [17]) in comparison with the respiration chamber or SF_6_ methods. There have been considerable inconsistencies in the precision and accuracy of estimating CH_4_ in cattle with the GFS [12,18]. Hammond et al. [12] reported that the GFS did not detect treatment and individual animal differences in CH_4_ production, whereas respiration chamber and SF_6_ techniques were able to detect these differences. To improve the estimates of CH_4_ emission in the GFS, the number, timing and duration of daily gas measurements relative to diurnal patterns of CH_4_ production need to be evaluated [12]. Although many studies have been conducted in large ruminants using GFS in comparison with respiration chamber and SF_6_ techniques, to our knowledge there are not yet published research results on CH_4_-emission estimation from goats using GFS developed for small ruminant use. Small ruminants are anticipated to play a crucial role in mitigating the impacts of future climate change and ensuring food security to meet demands of the growing global population [19]. We hypothesized that CH_4_-measurement frequencies at different times of the day may influence daily CH_4_-production estimates. Thus, the objectives of this study were to assess GFS against a stationary individual, open-circuit indirect, head-box animal calorimetry system (CS) [20,21] in measuring CH_4_ emission by growing Boer goat wethers and to determine the optimal frequency and timing of daily measurements using GFS. This head box or head hood respiration calorimetry system, which works in the same manner as indirect open-circuit respiration chambers, has been used at the American Institute for Goat Research for more than 20 years for the measurement of CH_4_ emission and heat production in small ruminants [20,21]. The goal of the research was to develop an accurate means of measuring CH_4_ emission under normal production conditions in order to develop management practices to minimize ruminant CH_4_ emission and heat energy loss from animals.

## 2. Materials and Methods

### 2.1. Animals and Procedures 

The study protocol was approved by the Langston University Animal Care and Use committee (Approval Number: 22-002; 7 January 2022). Sixteen Boer wethers, initially weighing 20 ± 0.71 kg and 7.4 ± 0.08 months of age, were used. Prior to the experiment, the animals were given a CD and T vaccine (against *Clostridium perfringens* type C and D and tetanus) and underwent training for accessing both Calan Gates (American Calan Inc., Northwood, NH, USA) and GFS. Moreover, a 3 wk acclimation period to the diet and other experimental conditions preceded commencement of the study.

All animals were housed in a group pen with dimensions of 12.2 m × 11.2 m, equipped with individual Calan Gate feeders. Oat hay (10% crude protein and 54% total digestible nutrients on dry matter basis based on information available in the Langston Interactive Nutrient Calculation system (LINC; http://40.65.112.141; last accessed on 2 February 2024) that was assumed adequate to meet nutrient requirements for maintenance was provided at approximately 3.2% of body weight daily on dry-matter basis at 900 and 1500 h in Calan Gate feeders. Additionally, a pelleted concentrate-based supplement (120 g/day) described by Tadesse et al. [22] and Hussein et al. [23] (about 17–18% crude protein on dry-matter basis) was delivered via GFS. Other nutrient contents in oat hay and supplement are given in Table 1. This supplement was used as a bait to attract animals to visit the unit regularly and also support body weight (BW) gain of approximately 75 g/day. Nutrient and energy requirements were calculated using the Supplemental Concentrate Needs Calculator of the Langston Interactive Nutrient Calculation system (LINC; http://40.65.112.141; accessed on 2 February 2024).

The experimental design was a replicated 4 × 4 Latin square with four periods each 14 days in length, four animal groups, and four gas-measurement protocols. At the beginning of the study, animals were grouped into four sets after stratification based on BW and randomly assigned to the four measurement protocols within each set (Table 2). Three of these protocols, denoted as GFS protocols (i.e., G-3T, G-4T and G-6T), involved potential gas measurements occurring three, four, and six times over a 24 h period, respectively, using GFS. The fourth protocol (CS) involved continuous gas measurements using a respiration calorimetry system (head-box type) with animals situated in metabolism crates. Gas measurements with GFS protocols were obtained each day of periods. Animals in the CS protocol also resided in the same group pen, but they were moved to the calorimetry room during the gas measurement days. Measurements with CS occurred continuously for 2 days in the first week and 2 days in the second week of each period. Body weight was recorded at the start and end of each period before the morning feeding and immediately before and after gas measurement in the calorimetry room. Feed intake was determined by weighing the offered feed and orts every morning in the Calan Gate Feeders and metabolic cages (for the CS protocol during measurement days).

### 2.2. Gas Measurement

#### 2.2.1. GreenFeed System

The GreenFeed system of C-Lock Inc. (Rapid City, SD, USA) was strategically positioned in the center of the group pen, allowing animals to access it freely, provided it was not in use by another animal. To regulate access, a narrow alley was constructed in front of the hood of GFS, limiting entry to one animal at a time. The detailed operation of GFS and the calculation of gas-emission rates have been outlined by Hammond et al. [12], Hristov et al. [24], and Huhtanen et al. [14]. In this study, non-dispersive infrared sensors (NDIR) for CH_4_ and CO_2_ and a paramagnetic O_2_ sensor for oxygen (O_2_) integrated into GFS were used for emissions of these gases while animals consumed the pelleted supplement within the hood. Although animals could visit any time, it necessarily did not result in gas measurement and feed drop. Gas measurements occurred only when an animal visited GFS during the designated time with pellet dispensation as a reward. Therefore, the visits to GFS were only considered as ‘valid’ visits when a feed bait was dropped and gas measurement occurred. Animal identification was facilitated by unique radio frequency identification (RFID) ear tags.

Animals subjected to G-3T, G-4T, and G-6T protocols received approximately 40, 30, and 20 g of a pelleted supplement (as fed basis), respectively, during each visit to GFS, resulting in a total daily supplement delivery of 120 g/animal for each treatment. The GFS, operated through an online interface via Wi-Fi connection, was programmed, controlling the time of supplement delivery with waiting periods of 8, 6, and 4 h for G-3T, G-4T and G-6T, respectively, during which time pellets were not dispensed and no gas measurements occurred. Also, GFS was remotely programmed to control the number of drops per feeding period, interval between drops and pellet quantity per drop.

Emissions of CH_4_ and CO_2_ consumption of O_2_ (g/day) were calculated from CH_4_, O_2_, and CO_2_ concentrations and air flow during visits to the feeder in the GFS, corrected by background concentrations, air flow, and calibration coefficients. To ensure measurement accuracy, GFS analyzers were auto-calibrated once daily using zero baseline gas (oxygen-free nitrogen) and a span gas mixture of nitrogen containing 5000 ppm CO_2_ and 1000 ppm CH_4_ (BOC Ltd., Manchester, UK). Calibration procedures were remotely controlled through GFS online interface. Carbon dioxide-recovery test was conducted one time with 4–5 repetitions per period by releasing a known amount of CO_2_ into the air intake manifold over a 3 min period. The air filter was cleaned or replaced weekly to eliminate dust and fine particulate materials that could impact the complete capture of exhaled gases. Data were regularly stored and ultimately downloaded through a web-based data management system provided by C-Lock Inc.

#### 2.2.2. Calorimetry System

The detailed procedures for gas measurement techniques using the CS have been elucidated by Puchala et al. [20,21]. A separate room with six metabolism crates fitted with head boxes of an indirect, open-circuit respiration CS (Sable Systems International, North Las Vegas, NV, USA) was used for determination of gas exchange and heat energy expenditure of sheep and goats. In each period, four goats being subjected to the CS protocol were moved to the CS room for gas measurements for a duration of 2 days in week 1 and 2 days in week 2 of the period by keeping them in an individual metabolism crate. Although the system affects animal behavior and is not representative of an open-air environment, it has been used as a standard method of gas measurement with livestock. In addition to the main diet (i.e., oat hay), animals received a similar amount of supplemental pellets (i.e., 120 g/day as two meals equally at 900 and 300 h) as animals on GFS protocols and water was offered twice daily as well.

Methane and CO_2_ concentrations were measured with infrared analyzers (CA-1B for CO_2_ and MA-1 for CH4; Sable Systems International), and O_2_ concentration was analyzed using a fuel cell FC-1B O_2_ analyzer (Sable Systems International). Prior to gas exchange measurements, analyzers underwent calibration with gases of known concentrations to ensure the validity and accuracy of flows. Ethanol combustion tests were conducted to verify the recovery of O_2_ and CO_2_ produced during measurements. The Brouwer [25] equation, excluding urinary nitrogen, was employed to calculate energy expenditure in GFS and CS based on O_2_ consumption and CO_2_ and CH_4_ production.

### 2.3. Samples and Laboratory Analysis

Feed and ort samples were collected twice a week on the same days when gas measurements were simultaneously carried out with CS and GFS. Composite samples were formed for each week, dried at 55 °C for 48 h in a forced-air oven, and kept frozen at −20 °C until analyses. The samples were ground to pass through a 1 mm screen and analyzed for dry matter (DM, ID No. 934.01), organic matter (OM, ID No. 942.05) [26], neutral detergent fiber inclusive of ash (NDF, ID No. 2002.04) with use of heat-stable amylase [27], and gross energy (GE) using a bomb calorimeter (Parr 6300; Parr Instrument Co., Inc., Moline, IL, USA) (Table 1). Fecal DM, ash, and GE contents were analyzed as described for feed and ort analysis. Fecal samples were collected for 4 days when animals were in the calorimetry room, but no feces were collected when they were with GFS to avoid potential effects on behavior of carrying fecal bags. Since all animals were fed a similar diet, digestible DM, OM and energy intakes of animals with the GFS were computed by multiplying the amount of DM, OM and energy ingested during the GFS period, with digestibility coefficients calculated for animals with the CS. 

### 2.4. Calculations and Statistical Analyses

Methane and CO_2_ emissions data (g/day) acquired from valid animal visits to GFS were extracted from the raw data for the final analysis. Methane yield (g/kg DM intake) and intensity (g/kg BW) were calculated by dividing CH_4_ production (g/day) by daily DM intake and BW, respectively. Methane and CO_2_ emissions, along with visit data, averaged for week were subjected to analysis using the Mixed Models Procedures of SAS [28]. Visit data were not applicable for the CS protocol. Data were analyzed following a Latin square design with the model: Yijkl = µ + αi + βj + (αβ)ij + γk + δl + eijk, where Yijk = the value of each individual observation, µ = overall mean, αi = effect of measurement protocol i, βj = effect of repeated measure of period l × week j except for digested nutrient intake (period l was the repeated measure), (αβ)ij = effect of the interaction between measurement protocol i and week j, γk = random effect of animal k, and eijk is residual.

Body weight, feed intake, digestion, and CH_4_ emission expressed in terms of g/kg organic matter (OM) and digestible OM (DOM) intakes, averaged over period, were analyzed by including measurement protocols as a fixed effect and animals within groups as random in the model. The least significant difference test was employed for comparing means of CS, G-3T, G-4T and G-6T. Statistically significant differences were acknowledged at *p* < 0.05, while values between 0.05 and 0.10 were considered tendencies for differences. Furthermore, Pearson correlation analysis was conducted between CH_4_ emissions estimated with GFS and CS using the same software [28], and scatter plots were generated to display the relationships.

## 3. Results

### 3.1. Body Weight, Feed Intake, and Digestion

Measurement protocols did not influence BW (*p* = 0.117), although noteworthy differences were observed in the consumption and digestion of total DM, OM and GE (*p* < 0.03) by goats (Table 3). Notably, GFS protocols (G-3T, G-4T, and G-6T) exhibited higher values compared with CS (*p* < 0.05). Although differences among protocols in hay intake were similar to those in total DM intake, this was not the case for pellet intake. The contributions of pellet intake to the total DM intake were 12.1, 12.2, 10.8 and 19.1% for G-3T, G-4T, G-6T, and CS, respectively, with the highest value for CS (*p* < 0.001) and a lower value for G-6T than for G-4T (*p* = 0.041). Digested DM, OM and GE were greater for the FGS than for CS (*p* < 0.05). 

### 3.2. Emission of CH_4_ and CO_2_

Methane production (g/day), yield (g/kg DMI) and intensity (g/kg BW) were affected by measurement protocols (*p* < 0.001; Table 4). Values were lower for CS than for G-3T, G-4T and G-6T (*p* < 0.05), with no significant differences observed among GFS protocols. Similarly, CH_4_ emission in g/kg DOM was lowest among protocols for CS *p* < 0.05), with no significant differences between G-3T, G-4T and G-6T. In line with emissions of CH_4_, average CO_2_ production (g/day) was significantly affected by measurement protocol, being lower for CS than for G-3T, G-4T and G-6T (*p* < 0.001). CO_2_ production for G-3T was higher than for G-6T (*p* < 0.05), with the values being intermediate for G-4T (*p* > 0.05). Energy expenditure in MJ/day and kJ/kg BW^0.75^ ranked G-3T > G-6T > CS and G-4T > CS (*p* < 0.05).

Table 5 presents the results of the correlation analysis between CH_4_ emission for CS and values for GFS protocols. Moderate to high correlations were observed between CS and average values of the three GFS protocols for CH_4_ production (r = 0.69; *p* = 0.004) and intensity (r = 0.75; *p* = 0.002). The corresponding correlation coefficients between CS and G-6T (r = 0.70 and 0.75; *p* < 0.004) and CS and G-4T (r = 0.67 and 0.76; *p* < 0.007) were significant and higher than between CS and G-3T (r = 0.62 and 0.59; *p* < 0.020). Conversely, the correlations between CS and GFS were generally low for CH_4_ yield and emission in g/kg DOM (*p* > 0.05) but only tended to be significant between CS and G-4T (*p* < 0.08). Figure 1 illustrates the relationship between the measurement protocols for CH_4_ emissions. According to the scatterplots, there was a positive linear relationship between CS and GFS for CH_4_ emissions, but the magnitude of relationship for CH_4_ production (g/day) and intensity (g/kg BW) were greater than for CH_4_ yield (g/kg DMI).

### 3.3. Number of Visits to GFS

Visit frequency per animal per day differed (*p* < 0.05) among the GFS protocols and ranked (*p* < 0.001) G-3T < G-4T < G-6T (Table 6). The duration of visits per animal per visit was similar between G-3T and G-4T (*p* > 0.05), but it was greater than for G-6T (*p* < 0.004). Considering the average daylight length from 0700 to 1900 h during the time of the study (i.e., March and April 2022), it was found that most visits occurred during the daytime (79%). The corresponding daytime and nighttime values for G-3T, G-4T and G-6T animals were 66% and 34%, 80% and 20%, and 90% and 10%, respectively.

## 4. Discussion 

### 4.1. Emissions of CH_4_ and CO_2_

Methane production as g/day was greater in the GFS than in the CS. Previous studies conducted with sheep [5,29] have shown DM intake to be highly positively correlated with CH_4_ production. In our study, the reduced DM intake of CS compared with that of GFS animals (8 and 15% lower for total DM and hay intakes, respectively) may partly be responsible for reduced CH_4_ emissions by CS animals. This is justified by the results of the correlation found between DM intake and CH_4_ production (g/day), which were moderate and positive (r = 0.58) for CS, but low and positive for GFS (r < 0.38). The reduced DM intakes of animals in respiration chambers have been reported in other studies (e.g., Bickell et al. [30]; Troy et al. [31]). Decreased substrate levels available for microbial fermentation in the rumen because of reduced DM intake may result in low CH_4_ emission in g/day from animals [6,32]. Correcting DM intake, CH_4_ yield (g/kg DM intake) was also lower in CS than in the GFS. In the literature, CH_4_ production in goats varies between 6.5 to 39 g/kg DM intake with a mean value of 18.4 g/kg DM intake [6]. In the present study, a CH_4_ yield of 19.3 g/kg DM intake in the CS is similar to this mean value, whereas a CH_4_ yield of 43.0 g/kg DM intake in the GFS was much greater than mean values in the literature [6]. 

Comparative analyses of the GFS with other techniques, particularly with respiration chambers, for estimating CH_4_ emission by cattle have been conducted (e.g., Hammond et al. [13]; Jonker et al. [16]; Rischewski et al. [33]) with inconsistent results. Factors that may contribute to such discrepancies include the independent standardization, calibration interval, calibration method, and standard gases used, which can lead to systematic differences between the methods [16]. In addition, Della Rosa et al. [9] noted that the main differences between the CS and the GFS in estimating CH_4_ emissions are related to differences in measurement duration and animal behavior when placed in the system. In our study, much lower CH_4_ emission for the CS compared with the GFS could involve variation in the frequency and timing of gas sampling. Notably, a majority of animal visits to the GFS occurred during the daytime, with relatively few visits at night. Unlike continuous CH_4_ measurement with the CS, measurement with the GFS was based on the time and number of animal visits to the GFS. This could potentially lead to an overestimation of daily CH_4_ emission, as visits at night tend to result in lower CH_4_ emission compared with visits during the daytime. In the present study, CH_4_ production in g/day estimated from nighttime visits was about 5.5% lower than that from daytime visits. This is supported by findings of Hammond et al. [12], where night/early morning visits by cattle were less frequent, contributing to an overestimation of daily CH_4_ emissions. 

In cattle, the GFS has reasonably estimated CH_4_ emission compared with the respiration chamber method [18,34]. No published research findings are available with goats using the GFS to justify the much higher CH_4_ emissions than with the CS. However, there have been two unpublished studies recently conducted at the American Institute for Goat Research (AIGR) of Langston University with objectives of determining the appropriate number of goats required for use of the GFS in confinement and grazing settings. The preliminary analysis of the data indicates that the CH_4_ production estimated with the GFS was also greater compared with the CS. This discrepancy in CH_4_ emission between these two systems might arise partly due to proportion of forage intake with greater proportion for the GFS (88%) compared with the CS (81%) because higher dietary forage to concentrate ratios favor CH_4_ production by ruminal microorganisms [35]. Also, additional factors such as a fast sampling rate, an accurate measurement of air flow rate and an NDIR detection limit may be assessed for the precise estimation of CH_4_ production in small ruminants. Particularly, the NDIR sensor may not be highly sensitive for detection of low concentrations of CH_4_ due to overlapping absorption spectra [36,37]. Therefore, the accuracy of the measurement of CH_4_ concentration in the background and in eructation samples may be problematic for small ruminants because of the low concentration of CH_4_ in breath/eructation samples, having an emission rate a tenth of that of cattle [34].

Similar CH_4_ emissions among GFS protocols were not expected because of considerable differences in the time and number of daily visits. Despite the number of nighttime visits being lowest for G-6T (10%) and highest for G-3T (34%), differences among GFS protocols for CH_4_ emissions in g/day, g/kg DMI, g/kg DOM and g/kg BW were only numerical. Conversely, Alemu et al. [38] reported the lowest CH_4_ emission when visits by beef heifers were the least. According to Jonker et al. [16], the timing, frequency and number of visits to a GFS unit could affect the power of each spot measure to predict the daily of CH_4_ emission estimates. Similarly, Waghorn et al. [39] indicated that the number of visits per animal is very important to derive reliable measures of daily CH_4_ emissions. 

Based on the correlation analysis conducted, there were moderate to strong relationships between CS and G-4T and CS and G-6T for CH_4_ production (g/day) and intensity (g/kg BW), and low but significant correlation coefficients between CS and G-3T. The reason for low relationships between CS and G-3T may involve the lesser number of daily visits to GFS and perhaps the distribution of times or specific hours of the day of the visits. The greater variability of CH_4_ emission (g/day) for the GFS (CV = 22.5–26.1%) than for the CS (CV = 17.5%) but a lesser variability of DM intake for all protocols is a likely explanation for the overall CH_4_ emission in g/kg DM intake being similar among protocols but correlations between the GFS and CS for CH_4_ emissions in g/kg DM intake being relatively low. Similarly, Hammond et al. [12] reported a lack of correlation between the GFS and respiration chamber estimates of CH_4_ emissions by cattle. Because there were moderate to high correlations for G-4T versus CS in the present study, the GFS system may potentially be useful for identifying treatment differences or ranking animals based on CH_4_ production.

### 4.2. Number of Visits to GFS

For accurate estimations of CH_4_ emissions, it is crucial to have sufficient visits to the GFS proportionally distributed over a 24 h period [40]. Although potential pellet feeding time/visits to GFS were equally spaced over the day in our study, G-3T animals had relatively higher numbers of nighttime visits compared with G-4T and G-6T. This is probably not surprising because animals in this protocol were permitted to visit the GFS only three times a day as opposed to four and six times a day for G-4T and G-6T, respectively. This suggests that increasing the frequency of potential daily visits/pellet consumption periods from three to six might have caused the animals to visit the GFS more frequently during the daytime than at night. The higher number of nighttime visits to the GFS by G-3T and G-6T animals in the present study may be attributed to the goats being more active during the daytime than nighttime. Although animals visited the GFS less than permitted, the average number of daily visits per animal (i.e., 2.03–3.75 visits) and length of duration per visit (i.e., 4.1–4.6 min) found in the present study agrees with recommendations based on research with cattle [41,42]. According to the authors, CH_4_ emissions become less accurate if the measurements or visits are infrequent (less than two visits per day) and if the duration is less than 3 min per visit.

## 5. Conclusions

In the present study, CH_4_ emissions measured from growing Boer wethers using the GFS were much higher than measurements with the CS. However, the results of the correlation analysis confirmed the presence of moderately strong to strong relationships between CS and GFS protocols for CH_4_ emissions in g/day and g/kg BW, with higher coefficients for G-4T and G-6T than for G-3T. The estimates of CH_4_ emission for the three GFS protocols were similar irrespective of differences in the time and number of daily visits. This indicates that the numbers and time of gas measurements used in the present experimental conditions are not highly influencing factors in estimating CH_4_ emission in goats. Additional factors need to be considered to elucidate the differences in CH_4_ emissions between the GFS and the CS. Our study suggests that to estimate CH_4_ emission using GFS in a confinement setting, a protocol should consist of at least four daily measurements, such as at 700–800, 1300–1400, 1900–2000 and 100–200 h, to potentially detect treatment differences and (or) for to rank animals for their CH_4_ emissions.

## Figures and Tables

**Figure 1 animals-14-00835-f001:**
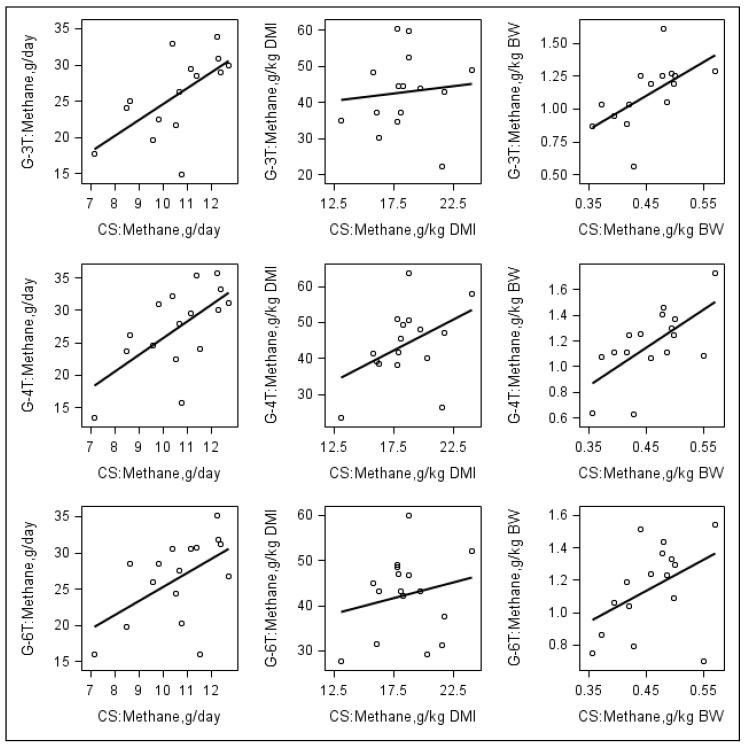
Relationship between methane emissions: g/day, g/kg dry matter intake, and g/kg body weight determined using GFS and CS. G-3T, G-4T, G-6T = Gas measurements taken three, four and six times a day using GreenFeed system, respectively; CS = Gas measurement taken throughout the day using calorimetry system; DMI = dry matter intake; BW = body weight.

**Table 1 animals-14-00835-t001:** Nutrient composition of oat hay and pelleted supplement fed to Boer goats.

Item	Oat Hay	Supplement
Dry matter, g/kg of diet	920	916
Organic matter, g/kg DM	904	913
Neutral detergent fiber, g/kg DM	606	260
Gross energy, MJ/kg DM	16.0	16.4

DM = Dry matter.

**Table 2 animals-14-00835-t002:** Methods, frequency and time of daily gas-measurement protocols employed in the study.

Protocol	Method of Measurement	Number of Measurements	Time of Measurement
G-3T	GreenFeed	3	0600–0700, 1400–1500, and 2200–2300 h
G-4T	GreenFeed	4	0700–0800, 1300–1400, 1900–2000, and 0100–0200 h
G-6T	GreenFeed	6	0800–0900, 1200–1300, 1600–1700, 2000–2100, 0000–0100, and 0400–0500 h
CS	Calorimetry	Continuous	Throughout the day

**Table 3 animals-14-00835-t003:** Effect of measurement protocol on body weight, feed intake and digestion.

Variable	Protocol ^1^					Week			*p*-Value	
	G-3T	G-4T	G-6T	CS	SEM	1	2	SEM	Protocol	Week
Body weight (kg)	23.3	23.4	23.2	23.3	0.48	23.0 ^a^	23.5 ^b^	0.47	0.117	0.004
Intakes (g/day)										
Total dry matter	614 ^b^	625 ^b^	635 ^b^	577 ^a^	13.9	596 ^a^	630 ^b^	12.1	0.001	0.001
Oat hay	540 ^b^	549 ^b^	566 ^b^	468 ^a^	13.6	515 ^a^	546 ^b^	11.5	<0.001	0.002
Pellet	74.2 ^bc^	75.8 ^c^	68.7 ^b^	110 ^a^	2.47	80.9	83.3	1.74	<0.001	0.344
Organic matter	556 ^b^	565 ^b^	570 ^b^	521 ^a^	14.1	544	552	12.6	0.022	0.235
Gross energy (MJ/day)	9.86 ^b^	10.0 ^b^	10.1 ^b^	9.25 ^a^	0.25	9.62	9.73	0.34	0.026	0.142
Digestion (g/day) ^2^										
Dry matter	359 ^b^	368 ^b^	374 ^b^	320 ^a^	15.9				0.023	
Organic matter	339 ^b^	348 ^b^	352 ^b^	304 ^a^	14.7				0.024	
Energy (MJ/day)	5.84 ^b^	6.01 ^b^	6.09 ^b^	5.24 ^a^	0.26				0.026	

^1^ G-3T, G-4T, G-6T = Gas measurements taken three, four and six times a day using GreenFeed system, respectively; CS = Gas measurement taken throughout the day using calorimetry system; SEM = standard error of the mean. ^2^ Digestion data were averaged for two weeks. ^a,b,c^ Means with different superscript letters in a row differ significantly (*p* < 0.05) among the measurement protocols.

**Table 4 animals-14-00835-t004:** Methane and carbon dioxide emissions in goats measured with different measurement protocols.

Variable	Protocol ^1^					Week			*p*-Value	
	G-3T	G-4T	G-6T	CS	SEM	1	2	SEM	Protocol	Week
CH_4_ emission										
g/day	25.6 ^b^	27.3 ^b^	26.7 ^b^	11.1 ^a^	1.11	22.7	22.6	0.99	<0.001	0.969
g/kg DMI	42.6 ^b^	43.9 ^b^	42.4 ^b^	19.3 ^a^	2.03	38.1	35.8	1.79	<0.001	0.105
g/kg BW	1.11 ^b^	1.18 ^b^	1.16 ^b^	0.49 ^a^	0.052	0.99	0.97	0.046	<0.001	0.447
g/kg OMI	46.4 ^b^	48.3 ^b^	46.7 ^b^	20.4 ^a^	2.20	39.4	41.5	3.14	<0.001	0.352
g/kg DOM ^2^	79.5 ^b^	80.2 ^b^	77.4 ^b^	35.7 ^a^	5.22				<0.001	
CO_2_ emission (g/day)	746 ^c^	727 ^bc^	712 ^b^	425 ^a^	23.2	650	654	21.8	<0.001	0.729
Heat production										
MJ/day	8.06 ^c^	7.81 ^bc^	7.61 ^b^	4.41 ^a^	0.271	6.98	6.96	0.253	<0.001	0.844
kJ/kg BW^0.75^	764 ^c^	737 ^bc^	720 ^b^	429 ^a^	21.2	669	657	19.3	<0.001	0.327

^1^ G-3T, G-4T, G-6T = Gas measurements taken three, four and six times a day using GreenFeed system, respectively; CS = Gas measurement taken throughout the day using calorimetry system; BW = body weight; DMI = dry matter intake; DOM = digestible organic matter; SEM = standard error of means. ^2^ Digestion data were averaged for two weeks. ^a,b,c^ Means with different superscript letters in a row differ significantly (*p* < 0.05) among the measurement protocols.

**Table 5 animals-14-00835-t005:** Correlations between measurement protocols for methane emissions from goats.

	CH_4_ (g/day)	CH_4_ (g/kg DMI)	CH_4_ (g/kg DOM)	CH_4_ (g/kg BW)
CS with G-3T ^1^	0.623 *	0.109	0.170	0.593 *
CS with G-4T ^1^	0.671 **	0.471	0.489	0.755 **
CS with G-6T ^1^	0.702 **	0.310	0.291	0.749 **
CS with GFS ^2^	0.693 **	0.321	0.332	0.745 **

^1^ G-3T, G-4T, G-6T = Gas measurements taken three, four and six times a day using GreenFeed system, respectively; CS = Gas measurement taken throughout the day using calorimetry system. DMI = dry matter intake; BW = body weight; DOM = digestible organic matter. ^2^ Correlation between CS and averages of three GFS measurement protocol. * *p* < 0.05; ** *p* < 0.01.

**Table 6 animals-14-00835-t006:** Number and duration of visits to the GFS by growing goats.

Variable	Protocol ^1^				Week			*p*-Value	
	G-3T	G-4T	G-6T	SEM	1	2	SEM	Protocol	Week
Number of visits per animal per day	2.03 ^a^	2.76 ^b^	3.75 ^c^	0.114	2.80	2.89	0.093	<0.001	0.517
Time spent (min) per animal per visit	4.54 ^b^	4.58 ^b^	4.07 ^a^	0.132	4.49	4.29	0.117	0.002	0.093

^1^ G-3T, G-4T, G-6T = Gas measurements taken three, four and six times a day using GreenFeed system, respectively; SEM = Standard error of mean. ^a,b,c^ Means with different superscript letters in a row differ significantly (*p* < 0.05) among the measurement protocols.

## Data Availability

Data are contained within the article.

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
