# Peer review of "Determining Appropriate Numbers and Times of Daily Measurements Using GreenFeed System to Estimate Ruminal Methane Emission of Meat Goats"

_animals, 2024, doi:10.3390/ani14060835_

Round 1
Reviewer 1 Report
Comments and Suggestions for Authors
This is potentially a very interesting paper on the measurement of methane emissions from goats using the GreenFeed system. The experimental design is appropriate and the results are presented in a few easily understandable tables and graphs.
However, I would like to ask the authors to thoroughly edit the text body of the paper and improve the written English in such a manner that it actually meets the standards of a scientific manuscript. Ways to do that is to simply cut out all unnecessary information, avoid repetition of information in the text that has been shown in the tables and graphs already, and considerably shortening drawn-out sentences. I am fully aware that the editing process in any language is painstaking and difficult, but I have had a hard time following the actually very good and interesting research results by repeatedly having to re-read sentences/full paragraphs to understand them.
I included a few comments in attached PDF as there were no line numbers in the manuscript.

Please see comments above (suggestions for authors)!
Author Response
Revision comments for Animals-2880841
Response to reviewer #1
Comment 1: regarding the need to improve the English language.
Response: We have made changes to improve the language.
Response 2: This sentence presented at the end of the simple summary is too long and quite confusing to read. Please split it up and rewrite in a more concise manner.
Response: A change has been made by omitting unnecessary phrases.
Comment 2: I think it is good to include quantitative results in the abstract, but you may omit a few details to make it more readable.
Response: The comment is somewhat unclear. The Abstract section does contain quantitative results. Also, we have added more quantitative results in the abstract.
Comment 3: You don't need to repeat 0700-0800, 1300-1400, 1900- 2000, and 0100-0200 h at the end of the abstract as you mentioned the time windows already earlier in the abstract.
Response: Thanks. The change has been made as per the suggestion.
Comment 4: In general, if you put this in plural (enteric emissions), you can omit the article. If you leave it in singular, you have to include an artilce (the) to make it flow better. So, in this case, simply state that "enteric CH4 emissions represent a sizeable energy loss..."
Response: The suggestion ("enteric CH4 emissions represent a sizeable energy loss...") has been adopted.
Comment 5: There needs to be a reference included in the use of calorimetry system, or in the Materials/Method section.
Response: References are already cited at the end of the Introduction section and beginning of the Materials and Methods section 2.2.2. We have now also cited those references in this sentence.
Comment 6: I think you can write the design of the study more succinctly: the four periods were your replications, if you used 16 goats then 2 per treatment; 4 groups of animals randomly assigned to each treatment (4) during each of the 4 periods. You will also have to mention that animals were stratified by weight. It might help to include a simple table with all these details, so you don't have to put in the text all the treatment details.
Response: Thanks for this suggestion. There were 4 animals (total 16; 4 animals/treatment and 4 treatments) per treatment in each period. A change has been made and a new table containing the details of the treatments has been inserted.
Comment 7: the first few sentences of the statistical analysis part can be moved to the respective sections or paragraphs in the material and methods part. You would add in the statistical analysis part that you analyzed data as Latin Square with 4 periods, that the 2 animals per group were averaged (I assume) if not entered as subsample etc....
Response: One sentence has been deleted, which was repetitive. Some sentences in this section are common for sections 2.2.1 and 2.2.1, and thus moving this part to the respective sections in the Materials and Methods section (sections 2.2.1 & 2.2.2) will create redundancy. To clarify the issue, we modified the heading as ‘Calculation and statistical analyses.’ However, some changes have been made.
Response to reviewer #2
Comment 1: A recommendation must be inserted at the end of the summary.
Response: A summary was provided, and it has been further modified.
Comment 2: Do not repeat the words contained in the text in the keywords.
Response: A change has been made.
Comment 3: In the introduction, authors must insert a sentence that addresses the main impacts that their research seeks to address, just as a hypothesis is necessary.
Response: Good suggestion. A sentence has been added at the end of the Introduction section as suggested and a hypothesis has been added.
Comment 4: In approach regarding the breed studied is necessary.
Response: The breed used in this study was the Boer goat and it has already been stated in the first paragraph of the Material and Methods section.
Comment 5: What is the objective of the research?
Response: The objective of the study has already been stated towards the end of the Introduction section.
Comment 6: I believe that images from the experiment could illustrate the analysis carried out.
Response: The comment is not clear, but the protocol of this study has been presented in a table now.
Comment 7: laboratory analysis (item 2.3) must be better detailed, and the numbers of the protocols followed in the analyzes must be inserted.
Response: Good suggestions. The feed analysis has been described now in greater detail, along with the method number.
Comment 7: The statistical models adopted must be inserted.
Response: A statistical model has been inserted as commented
Reviewer 2 Report
Comments and Suggestions for Authors
The study is well written and structured, I suggest small changes:
-A recommendation must be inserted at the end of the summary;
-Do not repeat the words contained in the text in the keywords.
-In the introduction, authors must insert a sentence that addresses the main impacts that their research seeks to address, just as a hypothesis is necessary.
-An approach regarding the breed studied is necessary
-What is the objective of the research?
-I believe that images from the experiment could illustrate the analyzes carried out.
-laboratory analysis (item 2.3) must be better detailed and the numbers of the protocols followed in the analyzes must be inserted.
-The statistical models adopted must be inserted
- Top of Form ??????
Author Response

(The authors gave the same response as above.)

Round 2
Reviewer 1 Report
Comments and Suggestions for Authors
I included just a few comments in the PDF.

See comments in the PDF. The paper has been improved substantially.
Author Response
All the English revisions suggested by the reviewers have been incorporated except the following: "To make it even more concise, you could say :"... the appropriate quantities of gas measurements". Here we did not actually measure the quantities of gas, Therefore, the term "numbers" is most appropriate.